# Chondrosarcoma of the Femur: Is Local Recurrence Influenced by the Presence of an Extraosseous Component?

**DOI:** 10.3390/cancers16020363

**Published:** 2024-01-15

**Authors:** Minna K. Laitinen, Michael C. Parry, Guy V. Morris, Robert J. Grimer, Vaiyapuri Sumathi, Jonathan D. Stevenson, Lee M. Jeys

**Affiliations:** 1Department of Orthopaedics and Traumatology, Helsinki University Hospital, University of Helsinki, 00100 Helsinki, Finland; 2Royal Orthopaedic Hospital, Birmingham and Aston University Medical School, Aston University, Birmingham B4 7ET, UK; michael.parry3@nhs.net (M.C.P.); guy.morris@nhs.net (G.V.M.); jonathan.stevenson@nhs.net (J.D.S.); 3Royal Orthopaedic Hospital, Birmingham B31 2AP, UK; rob.grimer@btopenworld.com (R.J.G.); vaiyapuri.sumathi@nhs.net (V.S.); 4Royal Orthopaedic Hospital, Faculty of Health Sciences, Aston University, Birmingham B4 7ET, UK; lee.jeys@nhs.net

**Keywords:** chondrosarcoma, margin of excision, femur, survival

## Abstract

**Simple Summary:**

This study focused on conventional central chondrosarcoma (CS) in the femur, investigating the impact of an extraosseous tumor component on local recurrence-free survival (LRFS) and disease-specific survival (DSS). Analyzing data from 202 patients, we identified factors influencing LRFS and DSS, highlighting the significance of the extraosseous tumor component, histological grade, and achieved surgical margins. Notably, the location of the extraosseous component within the femur significantly affected local recurrence rates, emphasizing the challenge of achieving adequate margins, especially in the region around the greater trochanter. The study underscores the importance of precise imaging and the need for aggressive resection to prevent local recurrence, particularly in cases involving the proximal extremity of the femur. Understanding these factors aids in tailoring surgical approaches for improved outcomes in managing femoral CS.

**Abstract:**

Background: Chondrosarcoma (CS) is the second most common surgically treated primary malignancy of the bone. The current study explored the effect of the margin and extraosseous tumor component in CS in the femur on local recurrence (LR), LR-free survival (LRFS), and disease-specific survival (DSS). Methods: Among 202 patients, 115 were in the proximal extremity of the femur, 4 in the corpus of the femur, and 83 in the distal extremity of femur; 105 patients had an extraosseous tumor component. Results: In the Kaplan–Meier analysis, factors significant for decreased LRFS were the extraosseous tumor component (*p* < 0.001), extraosseous tumor component arising from the superior aspect (*p* < 0.001), histological grade (*p* = 0.031), and narrow surgical margin < 3 mm (*p* < 0.001). Factors significantly affecting DSS were the histological grade (*p* < 0.001), extraosseous component (*p* < 0.001), LR (*p* < 0.001), metastases (*p* < 0.001), and surgical margin (*p* < 0.001). Conclusions: In CS of the femur, the presence of an extraosseous tumor component has a predictive role in LRFS, and extraosseous tumor component arising from the superior aspect was significant for decreased LRFS. Wide margins were more commonly achieved when the tumor had only an intraosseous component, and the rate of LR was significantly higher in cases with an extraosseous tumor component. When the extraosseous component arose from the superior aspect of the femur, LR occurred more frequently despite achieving adequate margins.

## 1. Introduction

Chondrosarcoma (CS) is the second most common surgically treated primary malignant bone tumor, accounting for approximately 20% of all bone sarcomas [1,2,3,4]. Conventional CSs constitute about 85% of all CSs and can be categorized according to their location within the bone, as central or (secondary) peripheral CS [5]. Central and peripheral CS are histologically similar and graded as low grade (grade 1) or high grade (grades 2 and 3). Grade 1 CS has more recently been classified as an atypical cartilaginous tumor (ACT) in the extremities, an intermediate tumor that is locally aggressive but does not metastasize [3]. When arising from the pelvis or axial skeleton, grade 1 CS behaves in a more aggressive manner and can metastasize [4,6,7,8,9].

Due to a relative insensitivity to chemotherapy and radiotherapy, CS is considered to be a surgical disease [4,10,11]. In managing CS, considering tumor grade and stage at diagnosis is vital for treatment planning and predicting the overall disease-specific survival (DSS) [12,13,14]. Additionally, achieving a sufficient surgical margin during resection significantly influences the likelihood of local recurrence (LR), playing a crucial role in long-term survival [15,16,17].

The anatomic location also predicts the behavior of the tumor; pelvic and axial CSs behave more aggressively than CS in an extremity, despite comparable grades. Acral CS and CS of the proximal humerus demonstrate improved overall survival compared with other locations in the extremities or pelvis [13]. The presence of a cortical breach and extraosseous tumor within the soft tissue has also been shown to have a major impact on survival [18,19].

Tumors arising from the femur, especially the proximal extremity of femur, raise specific challenges. The relation of the proximal extremity of the femur to the surrounding musculature, particularly the adductors of the hip and vasti muscle origin, means that tumors extending beyond the cortex can expand into the surrounding muscles without compartments to limit their growth [20]. The posterior aspect of the distal extremity of the femur also has no natural barriers, and extraosseous tumor components within the soft tissue make achieving a clear margin during resection even more challenging.

The aim of this study was to investigate the effect of an extraosseous tumor component of conventional central CS arising in the femur on LR and local recurrence-free survival (LRFS).

## 2. Materials and Methods

The study comprised a retrospective analysis of patients treated for central conventional CS in the femur at a singly tertiary sarcoma center—specifically, at the Royal Orthopaedic Hospital in Birmingham, UK—between January 1995 and October 2020. This review, approved by the institutional ethical review board, included patients diagnosed and treated at the hospital, excluding cases where patients were initially treated elsewhere or referred for managing recurrent tumors. Survivors were required to have at least 2 years follow-up, during which they were monitored for LR or metastases in line with European Society for Medical Oncology (ESMO) guidelines [21].

Details were collected on the clinical data and oncological outcomes, including LRFS. Primary surgery was defined as the method that concluded the first-line treatment. Resection specimens were examined by specialist bone sarcoma pathologists for grade, margin status, tumor location, and cortical breach in relation to the superior aspect of the greater trochanter or elsewhere within the specimen (Figure 1).

Grade was defined by internationally agreed upon standards and described according to the World Health Organization classification [1,9]. The diagnostic criteria governing histological grade included cellularity, nuclear size, the presence of an abundant hyaline cartilage matrix or mucomyxoid matrix, and the presence of mitosis. The highest grade seen in histology was then recorded, even when this higher grade comprised only a small number of cells. The margin was defined according to the Enneking classification as wide, marginal, or intralesional [22]. Because the Enneking definition of marginal and wide excision is inherently subjective and may vary depending on who is assessing the margin, we assessed the width of the surgical margin in millimeters. The histological status of the surgical margin was defined as follows: intralesional as microscopically positive (0 mm), marginal when margin was <3 mm, and wide when margin was ≥3 mm. The tumor was classified as proximal extremity of femur when the largest tumor mass was located in the subtrochanteric area, between the tip of the trochanter and 5 cm below the lesser trochanter. Tumors were classified as being located in the corpus of femur when the largest tumor mass was located in the diaphyseal area of thick cortical bone. Tumors were classified as being located in the distal extremity of femur when the biggest mass was located in the distal third of the femur.

The histological diagnosis and treatment plan were defined by a multidisciplinary team comprising specialist surgeons, radiologists, and pathologists. The primary outcome measure was LRFS. Secondary outcome measures were factors that influenced LR and DSS.

### Statistical Analysis

Patient survival rates were assessed using the Kaplan–Meier method with 95% confidence intervals (CIs) as median observation times to estimate the LRFS and DSS. DSS was defined as the time from diagnosis to disease-related death and was censored at the date of the latest follow-up examination or death due to other causes. LRFS was defined as the time from the surgical procedure to LR and was censored at the date of the latest follow-up visit or death. LR was defined as tumor relapse evidenced by radiological confirmation and subsequent histological confirmation from biopsy or by an interval increase of 1 cm in the size of abnormal lesions on sequential imaging. Age was normally distributed and tested using the Shapiro–Wilk test.

Continuous variables are reported as medians and ranges, and between-group differences were analyzed using the one-way Mann–Whitney test. The Pearson chi-squared test was used to compare variables between groups and the Mann–Whitney u-test for medians between groups. Univariate analysis was performed by comparing groups using the log-rank test with subsequent univariate and multivariate Cox proportional hazards analysis of continuous variables to identify predictors of LRFS and DSS.

The subdistribution hazard ratio (SHR) of the effect of LR on survival was calculated with a competing risk analysis. Synchronous metastases (metastases developed before LR, at the time of LR, or within 90 days after LR) and death due to other reasons were considered as competing events in analyses of the effect of LR on DSS. Statistical analyses were performed using SPSS Statistics 27.0 (IBM, New York, NY, USA) and STATA 17 (Stata, College Station, TX, USA). A *p*-value < 0.05 was considered significant.

## 3. Results

The final study population comprised 202 patients with conventional central CS in the femur who were identified from a database of 1034 primary CS patients (Figure 2).

Patient demographics are summarized in Table 1.

Information regarding the presence of an extraosseous tumor component was available for 189 patients (94%); 105 (56%) had evidence of an extraosseous component. Of these extraosseous components, 28 (36%) involved the superior aspect of the proximal extremity of the femur.

When looking at the margin status following resection in relation to the location of the extraosseous component, if the tumor had an extraosseous component in the superior aspect of the proximal extremity of the femur, the margin was intralesional in 18% (5/28), marginal in 54% (15/28), and wide in 29% (8/28). When the extraosseous component of the tumor arose elsewhere within the proximal extremity of the femur, not involving the greater trochanter, the margin achieved at resection was intralesional in 8% (4/49), marginal in 25% (12/49), and wide in 67% (33/49). In the distal extremity of femur, if the tumor had an extraosseous component, the margin was intralesional in 18% (4/22), marginal in 63% (12/19), and wide in 36% (13/36). There was a significant difference in margin status achieved at resection depending on the location of the extraosseous component (*p* = 0.005).

### 3.1. Predictors of LR and LRFS

The overall incidence of LR was 23% (47/202 patients). Margin as a factor affecting LR is summarized in Table 2. The extraosseous tumor component was a significant factor in LR (*p* = 0.005).

In grade 1 CS, the LRFS was 100% at 1 year, 93% (95% CI 85–100) at 3 years, and 88% (95% CI 79–97) at 5 and 10 years. In grade 2 CS, LRFS was 95% (95% CI 90–97) at 1 year, 81% (95% CI 73–89) at 3 years, and 70% (95% CI 60–81) at 5 and 10 years. In grade 3 CS, the LRFS was 82% (71–92) at 1 year and 69% (95% CI 56–82) at 3, 5, and 10 years.

In the Kaplan–Meier analysis, factors significant for LRFS were the extraosseous tumor component (*p* < 0.001), margin (*p* < 0.001), and histological grade (*p* = 0.031). The location of the extraosseous tumor component in relation to the superior aspect of the greater trochanter among the proximal extremity of the femur tumors was significant for LRFS (HR 7.4, 95% CI 3.0–18.2, *p* < 0.001; Figure 3). The phenomenon was similar in the distal extremity of the femur, with less significance (HR 3.1, 95% CI 1.0–8.9, *p* = 0.042).

In a multivariate Cox regression survival analysis, factors significant for LRFS were the extraosseous tumor component (HR 2.7; 95% CI 1.2–6.0, *p* = 0.021), histological grade (HR 2.9, 95% CI 1.7–5.2, *p* < 0.001), and margin achieved at resection (HR 0.2; 95% CI 0.11–0.33, *p* < 0.001).

### 3.2. Disease-Specific Survival

The overall disease-specific death rate was 18% (37/202). DSS was 100% at 1, 3, 5, and 10 years for grade 1 CS. For grade 2 CS, the DSS was 100% at 1 year, 81% (95% CI 72–89) at 3 years, 78% (95% CI 69–87) at 5 years, and 76% (95% CI 66–86) at 10 years. For grade 3 CS, the DSS was 92% (95% CI 84–99) at 1 year, 78% (95% CI 66–90) at 3 years, 68% (95% CI 53–83) at 5 years, and 50% (95% CI 28–72) at 10 years (Figure 4).

In a Kaplan–Meier analysis, factors significantly affecting DSS were the histological grade (*p* < 0.001), extraosseous component (*p* < 0.001), LR (*p* < 0.001), metastases (*p* < 0.001), and surgical margin (*p* < 0.001). In a Cox regression analysis, factors affecting DSS were increasing size (HR 1.060, 95% CI 1.023–1.097, *p* < 0.001) and increasing age (HR 1.034, 95% CI 1.012–1.055, *p* = 0.002). Following a competing risk analysis, factors affecting disease-specific failure were the tumor grade (SHR 3.3, 95% CI 2.2–4.9, *p* < 0.001), extraosseous tumor component (SHR 1.1e07, *p* < 0.001; Figure 5), and LR (SHR 6.1, 95% CI 3.1–11.8, *p* < 0.001).

## 4. Discussion

The femur is a common location for primary malignant and benign bone tumors [23]. In this study, 21% of the CSs in the extremities and pelvis were in the femur, making the femoral location the second most common location after the pelvis. In opposition to other primary bone sarcomas, the proximal extremity of the femur was a more common location for CS than the distal extremity of the femur. Survival of the patients with proximal extremity of femur CS was less favorable than that of patients with a tumor in the distal extremity of the femur. This is most likely because tumors in the proximal extremity of the femur were more often of higher grade and large in size at the time of diagnosis. In femoral incidental cartilage tumor findings reported in the literature, the site of the origin in the femur affects the final diagnosis, as the ratio of benign to malignant is 11:1 in the distal extremity of the femur and 1:1.5 in the proximal extremity of the femur [24]. Therefore, any cartilage lesions in the proximal extremity of femur, especially if suspicious for malignant CS, should be reviewed prudently.

LR is closely related to the adequacy of the resection margin and plays a significant role in disease-free survival [15,25,26]. The extraosseous component has also been shown to be important in the local and systemic control of CS [13,19,26]. However, in the most recent work by Welling et al., the authors chose to include peripheral and dedifferentiated CSs, making their results difficult to interpret [19]. By definition, peripheral CSs are always associated with an extraosseous component; in dedifferentiated CS, the tumor presents as a low-grade, cartilaginous tumor juxtaposed to a generally high-grade, mesenchymal, non-cartilaginous component. In dedifferentiated CS, the high-grade non-cartilaginous component is thought to be more responsible for the diminished survival [27,28,29,30,31].

Could the effect of local control be the result of the difficulty in achieving an adequate margin when there is an extraosseous component? In this study, as well as in general, an adequate margin was more commonly achieved when the tumor only had an intraosseous component. The rate of LR was significantly higher in cases with an extraosseous tumor component. We specifically wanted to study the effect of an extraosseous component in the superior aspect of the proximal extremity of the femur, in the region of the greater trochanter, where the tendons for the gluteus minimus, gluteus medius, and piriformis are attached to the bone and sufficient soft tissue coverage can be difficult to achieve. Our results demonstrate a higher rate of LR in cases in which the extracortical breach of the tumor occurred in the superior aspect of the proximal extremity of the femur, which one could argue is the result of a higher incidence of an inadequate margin in this area. The importance of the abductors for hip stability may tempt the surgeon to preserve as much of the abductors as possible in the resection in an attempt to improve the function and stability of the hip [32,33,34]. However, this may compromise the margin and increase the rate of LR. In cases in which the tumor breaches the cortex in the superior aspect of the proximal extremity of the femur, the principle aim of the operation, as with all other locations, is to remove the tumor in its entirety with a sufficient soft tissue margin to minimize the risk of LR, even if this has a detrimental effect on function. However, our results have shown that, despite the margin achieved at resection, the rate of LR was affected by the location of the extraosseous component with the margin not having a significant effect on LR in cases with an extraosseous component in the superior aspect of the greater trochanter. In the distal extremity of femur, the phenomenon was similar but less significant.

This study has some limitations, including those inherent to its retrospective design, which must be acknowledged. Even though this is the largest study to date on conventional central CS of the femur, the small number of patients included in each group may have influenced the results because statistical significance was difficult to achieve. This will have influenced the duration of recruitment and follow-up period for the study. However, this remains the largest study yet concerning conventional central femur CS in which only grades 1, 2, and 3 were combined. As CS is commonly seen in elderly patients, death due to other causes frequently occurs. Therefore, our statistical method of using survival data calculations with competing risk analysis gives a more accurate assessment of overall survival, which differs from methods reported elsewhere in the literature.

## 5. Conclusions

In conclusion, in conventional central CS of the femur, the presence of an extraosseous tumor component has a predictive role in LRFS. Our results have shown the value of accurate, up-to-date imaging when managing CSs of the femur. Despite the outcomes seeming independent of margins achieved during resection, given the potential difficulty in accurately histologically assessing the soft tissue margins in this location, the presence of an extraosseous component should alert the treating surgeon to the need for a more aggressive resection in order to not break the first rule of oncology surgery and not compromise on margins. This is particularly important when dealing with tumors erupting from the greater trochanter, where the margin must not be compromised in an attempt to preserve function. CSs in the proximal extremity of the femur behave more aggressively because they are more often of higher grade and greater in size, which highlights the need for aggressive surgery.

## Figures and Tables

**Figure 1 cancers-16-00363-f001:**
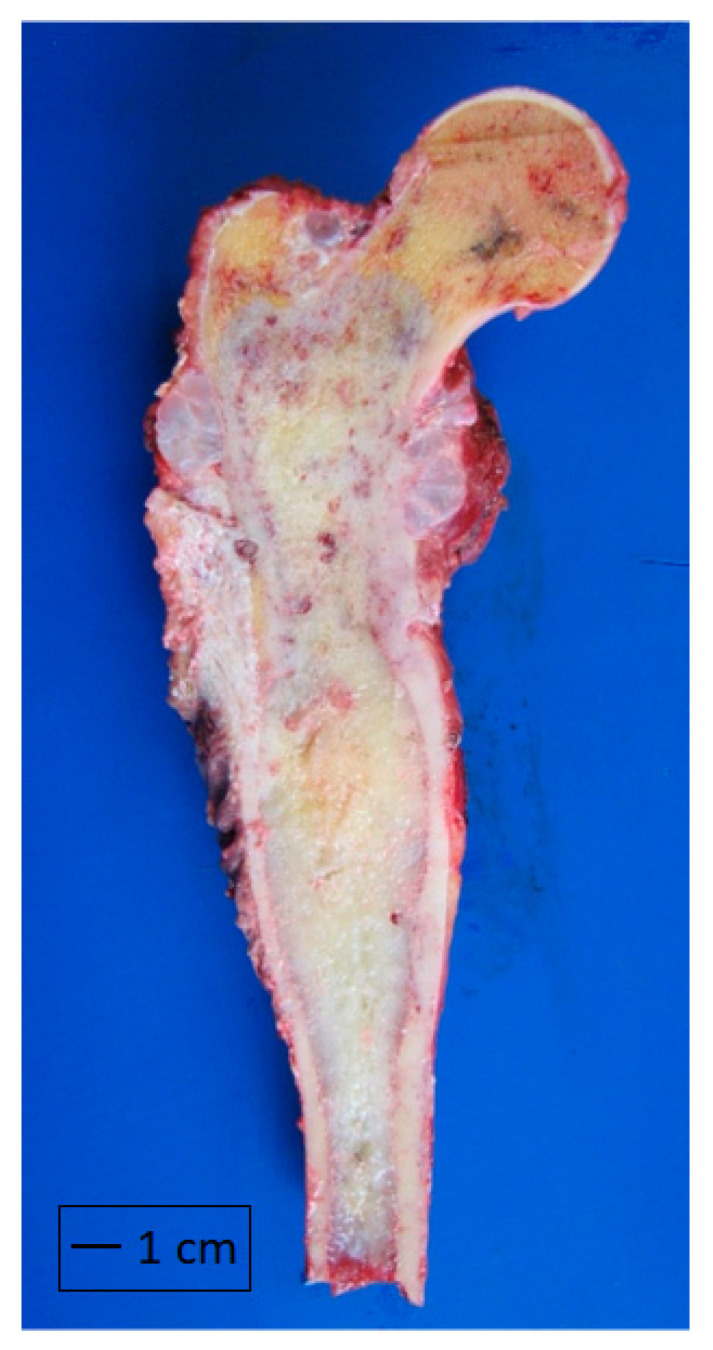
Resection specimen showing an extraosseous tumor component in both the metaphyseal area and the superior aspect of the greater trochanter.

**Figure 2 cancers-16-00363-f002:**
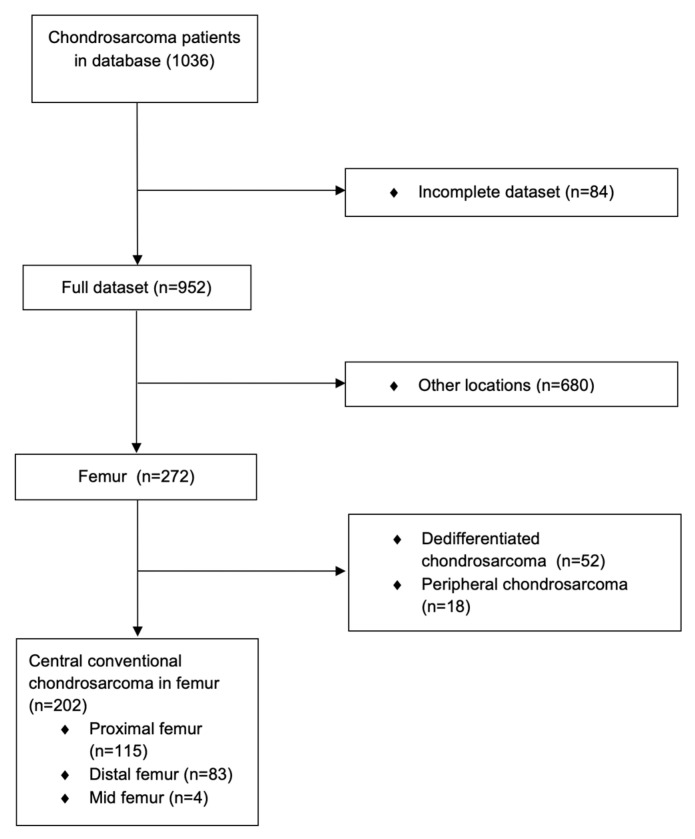
Flowchart of identification of patients with conventional central chondrosarcomas of the femur.

**Figure 3 cancers-16-00363-f003:**
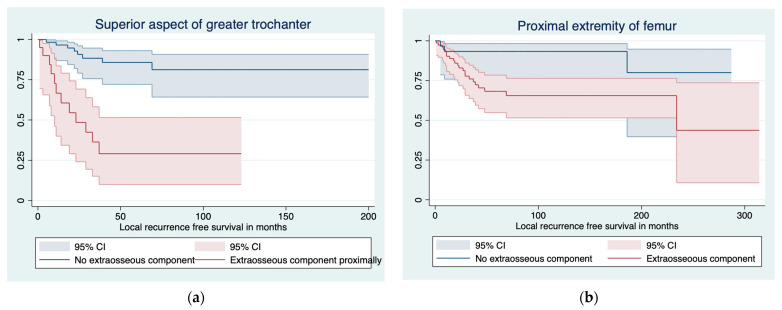
Local recurrence-free survival stratified by extraosseous tumor component in (**a**) the superior aspect of greater trochanter, (**b**) proximal extremity of femur, (**c**) distal extremity of femur, and (**d**) total femur.

**Figure 4 cancers-16-00363-f004:**
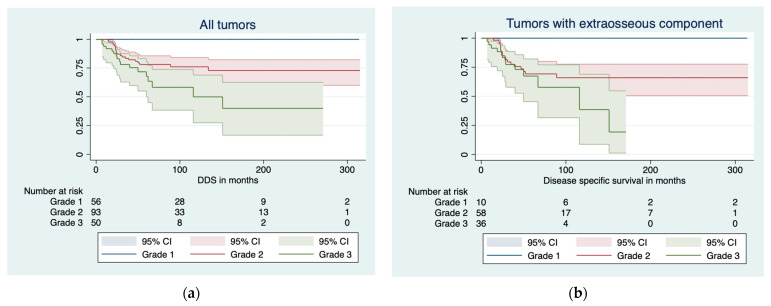
Disease-specific survival stratified by grade (**a**) in all tumors and (**b**) in tumors with extraosseous tumor component.

**Figure 5 cancers-16-00363-f005:**
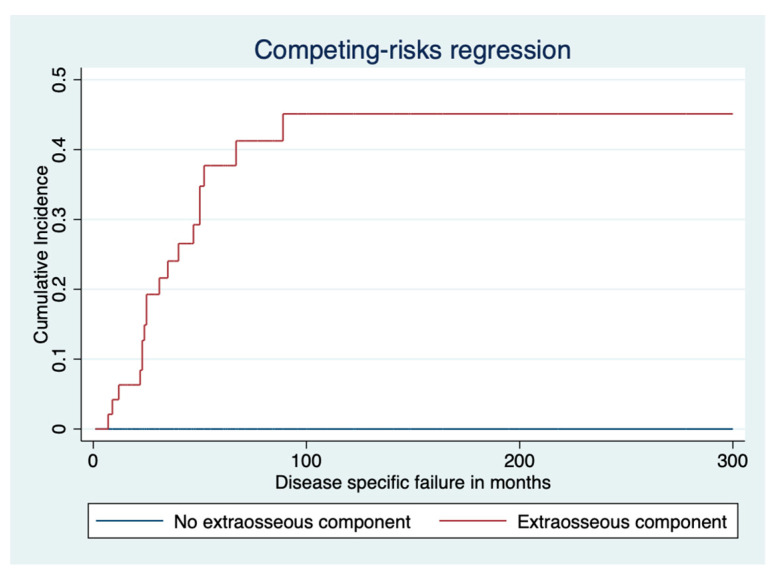
Disease-specific failure stratified by extraosseous tumor component in competing risk model.

**Table 1 cancers-16-00363-t001:** Characteristics of 202 central chondrosarcoma in femur cases (values are presented in number of cases).

Characteristics	Total	Grade 1	Grade 2	Grade 3
Eligible cases	202	57 (28%)	94 (46%)	51 (26%)
Site				
Proximal extremity	115 (57%)	20 (17%)	60 (52%)	35 (30%)
Distal extremity	83 (41%)	36 (43%)	32 (38%)	15 (19%)
Corpus	4 (1.9%)	1 (25%)	2 (50%)	1 (25%)
Male sex	110 (55%)	27 (25%)	50 (46%)	33 (30%)
Proximal extremity	65 (57%)	10 (15%)	33 (51%)	22 (34%)
Distal extremity	43 (52%)	17 (40%)	16 (37%)	10 (23%)
Corpus	2 (50%)	-	1 (100%)	1 (100%)
Age at surgery, median (range)	55 (8–95)	46 (10–85)	59 (8–95)	59 (23–88)
Proximal extremity	59 (8–95)	50 (10–85)	60 (8–85)	62 (25–87)
Distal extremity	51 (16–90)	44 (16–85)	57 (28–90)	54 (23–88)
Corpus	53 (28–81)	28	64 (47–81)	55
Mean tumor size, cm (range)	12 (1.5–46)	7.8 (1.5–46)	12 (3.0–30)	16 (4–40)
Proximal extremity	13 (2.0–40)	7.3 (2.0–17)	13 (3.5–30)	17 (6.0–40)
Distal extremity	9.3 (1.5–60)	8.0 (1.5–46)	10 (3.0–25)	11 (4.0–40)
Corpus	13 (6.0–19)	6.0	17 (16–19)	13
Extraosseous component	105 (52%)	10 (10%)	58 (55%)	37 (35%)
Proximal extremity	77 (67%)	6 (8%%)	42 (55%)	28 (37%)
Distal extremity	28 (38%)	4 (14%)	16 (57%)	8 (29%)
Corpus	1 (25%)	-	-	1 (100%)
Extraosseous component in cranial part of proximal extremity of femur	28 (24%)	1 (5%)	17 (28%)	10 (29%)
Median follow-up, months (range)	90 (0–315)	114 (0–314)	92 (0–315)	58 (0–271)
Proximal extremity	84 (0–315)	117 (13–314)	93 (0–315)	50 (0–182)
Distal extremity	100 (0–273)	114 (0–264)	93 (8–273)	78 (7–271)
Corpus	55 (10–104)	78	57 (10–104)	29
Pathologic fracture	20 (10%)	1 (1.8%)	12 (14%)	7 (14%)
Proximal extremity	13 (11%)	-	10 (17%)	3 (8.6%)
Distal extremity	6 (8.0%)	1 (2.9%)	2 (7.4%)	3 (21%)
Corpus	1 (25%)	-	-	1 (100%)
Metastasis	39 (19%)	-	21 (22%)	18 (35%)
Proximal extremity	26 (23%)	-	14 (23%)	12 (34%)
Distal extremity	12 (14%)	-	7 (22%)	5 (33%)
Corpus	1	-	-	1
Median time to metastasis in months (range)	22 (0–189)	-	26 (0–189)	16 (4–41)
Proximal extremity	19 (0–73)	-	20 (0–73)	18 (6–41)
Distal extremity	27 (0–189)	-	38 (0–189)	11 (4–16)
Corpus	21	-	-	21
Local recurrence	47 (23%)	7 (12%)	25 (27%)	15 (29%)
Proximal extremity	31 (27%)	2 (10%)	20 (33%)	9 (26%)
Distal extremity	15 (18%)	5 (11%)	5 (16%)	5 (33%)
Corpus	1 (25%)	-	-	1 (100%)
Median time to LR in months (range)	33 (0–234)	54 (14–186)	39 (0–234)	12 (1–22)
Proximal extremity	36 (0–234)	113 (39–186)	40 (0–234)	13 (1–22)
Distal extremity	28 (2–97)	35 (14–97)	38 (15–56)	8 (2–18)
Corpus	21	-	-	21
Dead for disease	37 (18%)	-	20 (21%)	17 (33%)
Proximal extremity	25 (22%)	-	14 (23%)	11 (31%)
Distal extremity	11 (13%)	-	6 (19%)	5 (33%)
Corpus	1 (25%)	-	-	1 (100%)

**Table 2 cancers-16-00363-t002:** Role of surgery in local recurrence of 202 central chondrosarcoma in femur cases (values are presented in number of cases (percentage) with number and percentage of local recurrences in brackets).

	Total	Grade 1	Grade 2	Grade 3
Curettage				
Proximal extremity	9 (8%) [5, 45%]	9 (45%) [2, 22%]	-	-
Distal extremity	18 (22%) [2, 11%]	16 (44%) [4, 25%]	2 (6%) [1, 50%]	-
Corpus	-	-	-	-
Resection				
Proximal extremity	96 (85%) [29, 30%]	11 (55%) [0]	58 (97%) [20, 34%]	27 (82%) [9, 33%]
Distal extremity	58 (70%) [8, 32%]	20 (56%) [1, 5%]	27 (84%) [4, 15%]	11 (73%) [3, 27%]
Corpus	4 (100%) [1]	1 (100%) [0]	2 (100%) [0]	1 (100%) [1, 100%]
Amputation				
Proximal extremity	6 (5%) [0]	-	2 (3%) [0]	4 (12%) [0]
Distal extremity	7 (8%) [2, 35%]	-	3 (9%) [0]	4 (27%) [2, 50%]
Corpus	-	-	-	-
Hindquarter				
Proximal extremity	2 (2%) [0]	-	-	2 (6%) [0]
Distal extremity	-	-	-	-
Corpus	-	-	-	-

## Data Availability

Deidentified patient data are available from the authors.

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
