# Peer review of "Chondrosarcoma of the Femur: Is Local Recurrence Influenced by the Presence of an Extraosseous Component?"

_cancers, 2024, doi:10.3390/cancers16020363_

Round 1
Reviewer 1 Report
Comments and Suggestions for Authors
The manuscript submitted for review is a very good example of retrospective study. The presented research provides a lot of valuable information. However, as a reviewer, I have to point out a few minor flaws. First of all, on Figure 1 lack of scale bar, which significantly disturbs the reading of the size of the presented pathological change. The text also contains minor editorial errors, such as: Chondrosarcoma(CS) – lack of space
It would also be necessary to change proximal, distal and mid femur to proximal/distal extremity of femur and corpus of femur, because used names are not consistent with the anatomical nomenclature.Author Response
The manuscript submitted for review is a very good example of retrospective study. The presented research provides a lot of valuable information. However, as a reviewer, I have to point out a few minor flaws. First of all, on Figure 1 lack of scale bar, which significantly disturbs the reading of the size of the presented pathological change.
A scale bar has been added on Figure 1.
The text also contains minor editorial errors, such as: Chondrosarcoma(CS) – lack of space
The spaces have been added and checked throughout the entire text.
It would also be necessary to change proximal, distal and mid femur to proximal/distal extremity of femur and corpus of femur, because used names are not consistent with the anatomical nomenclature.
Changed as suggested.
Reviewer 2 Report
Comments and Suggestions for Authors
The manuscript is very interesting and related to a relevant topic. The study is well conducted. Materials and Methods are clear and results detailed. The tables and charts are clarifying. The references are appropriate. It would be interesting to know if all the patients received a biopsy before the surgical treatment or in some cases a radiologic diagnosis was enough to decide the treatment
Author Response
The manuscript is very interesting and related to a relevant topic. The study is well conducted. Materials and Methods are clear and results detailed. The tables and charts are clarifying. The references are appropriate. It would be interesting to know if all the patients received a biopsy before the surgical treatment or in some cases a radiologic diagnosis was enough to decide the treatment
We appreciate the reviewer's insightful comment. However, our dataset lacked complete information on preoperative biopsies—specifically, details on whether a biopsy was conducted and its results—except for 160 patients. Consequently, we opted against reporting these findings. Notably, within this subset of 160 patients, only 8 underwent surgery without a preoperative biopsy. Of concern, 30 out of the 160 preoperative biopsies yielded reports of benign results or an inability to detect malignant cells.
Reviewer 3 Report
Comments and Suggestions for Authors
Please see the attached file.

Author Response
This is an intriguing article, particularly for clinicians. Below are some suggestions:
- Line 96: "was the" – I assume it should be "was then"?
Exactly-we thank you for this
- Line 102-107: How do you define the boundary between the mid and distal femur?
The mid femur/ corpus of the femur was defined as diaphyseal area of thick cortical bone. This has now been written for clarity.
- Line 116: Please define what MFS is. Avoid abbreviation.
MFS has been deleted unnecessary
- Line 117: Regarding the radiological diagnosis of metastasis, please describe the protocol for radiological follow-up (X-rays, CTs). How do you avoid surveillance bias?
The surveillance was done according to ESMO/European sarcoma network working group protocol. This has now been written for clarity and reference added. Bone sarcomas: ESMO clinical practice guidelines for diagnosis, treatment and follow-up. Ann Oncol. 2014; 25 Suppl 113-23.
- Line 120: How much increase in size will be considered significant?
This is a very good question and we don't know the absolute right answer for it. An increase of 1 cm will surely cause a reaction. We have added this for clarity.
- How do you define an "incomplete dataset"? Should some of them be analyzed as censored data? 7. Line 151-160: Is it possible to present results by surgical margin in mm instead of categorically as "intralesion, marginal, and wide" for the three locations?
We thank the reviewer for this comment. We did calculate the results in millimeters, but an increasing margin in millimeters did not show to be a good way to present the results, as LRFS does not improve significantly and linearly with increasing margin. Therefore, we would like to present it the way we have.
- Figure 3: If the authors want to specify the impact of the location of the extraosseous component, they may consider adding two additional curves – one for the extraosseous component in the middle femur and one for the extraosseous component in the distal femur. What about a merged curve for the total extraosseous component?
We thank the reviewer for this comment. We do have curves from every location including a merged curve. Since mid femur had only four cases with one extraosseous component, we suggest making a panel of four curves as follows. A) superior aspect of proximal femur b) proximal femur, c) distal femur and d) merged curve.
This panel of figures has now been added
- Figure 4: It would be interesting to know in which grade the extraosseous component exerts the highest impact.
We thank the reviewer for this comment, which indeed is very interesting. We have added a figure to show that grades 2 and 3 behave in a very similar way in tumors with extraosseous components.
Round 2
Reviewer 3 Report
Comments and Suggestions for Authors
Please see the attached file.

Author Response
- Line 87: Please include the full name of ESMO in its initial mention.
Has been added accordingly
- 2. Line 47: Reference 1 is outdated. Consider the following alternative references published in 2023: https://www.mdpi.com/2072-6694/15/9/2556 https://www.mdpi.com/2072-6694/15/14/3603 https://www.mdpi.com/2072-6694/15/7/1962
We agree the reference is outdated. More recent chondrosarcoma paper changed.
- 3. Line 99: Reference 3 is also outdated. Consider the following alternative references published in 2023: https://www.mdpi.com/2072-6694/15/6/1703
The reference has been added as well.